# The ZZ-type zinc finger of ZZZ3 modulates the ATAC complex-mediated histone acetylation and gene activation

Wenyi Mi [1,2], Yi Zhang [3], Jie Lyu [4], Xiaolu Wang[1,2], Qiong Tong[3], Danni Peng[1], Yongming Xue [1], Adam H. Tencer[3], Hong Wen[1,2], Wei Li [4], Tatiana G. Kutateladze [3] & Xiaobing Shi [1,2]

Recognition of histones by epigenetic readers is a fundamental mechanism for the regulation of chromatin and transcription. Most reader modules target specific post-translational modifications on histones. Here, we report the identification of a reader of histone H3, the ZZ-type zinc finger (ZZ) domain of ZZZ3, a subunit of the Ada-two-A-containing (ATAC) histone acetyltransferase complex. The solution NMR structure of the ZZ in complex with the H3 peptide reveals a unique binding mechanism involving caging of the N-terminal Alanine 1 of histone H3 in an acidic cavity of the ZZ domain, indicating a specific recognition of H3 versus other histones. Depletion of ZZZ3 or disruption of the ZZ-H3 interaction dampens ATAC-dependent promoter histone H3K9 acetylation and target gene expression. Overall, our study identifies the ZZ domain of ZZZ3 as a histone H3 reader that is required for the ATAC complex-mediated maintenance of histone acetylation and gene activation.

---

[1] Department of Epigenetics and Molecular Carcinogenesis, Center for Cancer Epigenetics, The University of Texas MD Anderson Cancer Center, Houston, Texas 77030, USA. [2] Center for Epigenetics, Van Andel Research Institute, Grand Rapids, Michigan 49503, USA. [3] Department of Pharmacology, University of Colorado School of Medicine, Aurora, Colorado 80045, USA. [4] Dan L. Duncan Cancer Center, Department of Molecular and Cellular Biology, Baylor College of Medicine, Houston, Texas 77030, USA. These authors contributed equally: Wenyi Mi, Yi Zhang and Jie Lyu. Correspondence and requests for materials should be addressed to T.G.K. (email: tatiana.kutateladze@ucdenver.edu) or to X.S. (email: xiaobing.shi@vai.org)

Post-translational modifications (PTMs) of histones, such as acetylation and methylation, play an important role in modulating chromatin structure and all chromatin-associated processes, including gene transcription[1,2]. Acetylation on histones is normally associated with gene activation, whereas methylation is linked to either gene activation or repression, depending on the specific residues being modified. In general, methylation on histone H3K4, H3K36, and H3K79 is enriched in actively transcribed genes, and di- and trimethylation on H3K9, H3K27, and H4K20 occupies silent genes[3]. The principal function of histone acetylation and methylation is to recruit or repel "reader" proteins that recognize the small moieties on modified histone residues and transduce the epigenetic signals at chromatin to distinct biological outcomes[4]. Studies from many laboratories have identified a number of readers selective to specific histone modifications, for example, the chromodomain (CHD) and the plant homeodomain (PHD) zinc finger recognizes methylation on distinct lysine residues of histones[5–8], whereas the bromodomain (BRD) and the YEATS domain bind to acetylated histones[9,10]. To date, almost all known histone readers are specific or sensitive to certain histone PTMs; however, a versatile reader that can accommodate common modifications on histones has yet to be discovered.

ZZZ3 (zinc finger ZZ-type containing 3) is a core subunit of the ATAC complex, a conserved metazoan histone acetyl-transferase (HAT) complex[11,12]. In mammals, ATAC contains one of the two highly related, mutually exclusive catalytic subunits, GCN5 or PCAF, which are shared with another major HAT complex, the Spt-Ada-Gcn5-Acetyltransferase (SAGA) complex[13,14]. While biological activities of the SAGA components have been extensively studied[15,16], the functions of most of the ATAC complex subunits, including ZZZ3, remain largely unknown.

Here, we report the identification of the ZZ-type zinc finger (ZZ) as a family of histone H3 reader. The solution NMR structure of the ZZ domain of ZZZ3 in complex with the H3 peptide reveals a unique binding mechanism involving caging of the N-terminal Alanine 1 of histone H3 in an acidic cavity of the ZZ domain. The ZZ domain specifically binds to histone H3 tail, making contacts mainly with the first four amino acids of histone H3. Surprisingly, this recognition is insensitive to methylation on H3K4 or H3R2, whereas acetylation on H3K4 moderately enhances the binding. The recognition of histone H3 by ZZ is essential for chromatin occupancy of ZZZ3 and functions of the ATAC complex in cells. Depletion of ZZZ3 or disruption of the ZZ-H3 interaction dampens ATAC-dependent promoter histone H3K9 acetylation and expression of target genes, including the ribosomal protein encoding genes. Overall, our study identified the ZZ domain of ZZZ3 as a versatile histone H3 reader that is required for the ATAC complex-mediated maintenance of histone acetylation and gene activation .

## Results

### The ZZ domain is a histone H3-recognizing module. 
ZZZ3 contains two conserved domains, a putative DNA-binding Swi3-Ada2-NCOR-TAFIIB (SANT) domain and a ZZ of unknown function (Fig. 1a). Screening for novel readers by a homemade histone peptide array, we found that the ZZ of human ZZZ3 binds to the N-terminal tail of histone H3 in a methylation-independent manner (Fig. 1b). In vitro histone-binding experiments (Fig. 1c) and histone peptide pulldowns (Fig. 1d and Supplementary Fig. 1a) further demonstrated that ZZZ3 ZZ specifically recognizes the H3 tail, but not other regions of H3, nor other histones. The N-terminal eight amino acids of H3 were sufficient for ZZZ3 binding; removal of the first four amino acids

of the H3 tail completely abolished the binding (Fig. 1e), suggesting that the very N-terminal end of H3 is critical for the ZZZ3-H3 interaction. Unlike other known H3-binding modules, such as the WD40 repeats of WDR5 and the first PHD finger (PHD1) of RBP2 that are sensitive to methylation on H3R2 and H3K4, respectively[17–19], the ZZZ3 ZZ-H3 interaction was not affected by methylation on these residues (Fig. 1d–f). In contrast, acetylation on H3K4 moderately enhanced ZZZ3 ZZ binding to H3 (Fig. 1g and Supplementary Fig. 1b), implying a novel mode of H3 recognition.

To determine whether H3 binding is a common feature of ZZ domains, we cloned 16 zinc fingers of all the human ZZ proteins that reside in, or shuttle into the nucleus[13], and tested their H3 binding activity in pull-down experiments with histone peptides and full-length histones. We found that 9 of the 16 ZZ domains could bind to histone H3 tail (Supplementary Fig. 1c–e), suggesting that H3 binding has evolved as a common function of ZZ domains within a subset of the protein family (Fig. 1h).

### Molecular basis of recognition of H3 by the ZZZ3 ZZ domain.
To define the molecular mechanism by which ZZZ3 recognizes histone H3, we determined the solution structure of the ZZZ3 ZZ domain (aa 816–874) in complex with an unmodified histone H3 tail peptide (aa 1–12) by NMR spectroscopy. The structural ensemble data were well defined throughout the protein and the four N-terminal residues of the H3 peptide (Supplementary Fig. 2a and Supplementary Table 1). The structure shows a cross-brace topology fold, with a central three-stranded antiparallel β-sheet (aa 847–849, 835–838, and 869–871), a short α-helix (aa 851–854), and a small two-stranded β-sheet (aa 817–818 and 831–832) at the N-terminus (Fig. 2a). The histone peptide is bound in an extended conformation, forming a short two-stranded antiparallel β-sheet with the residues 821–823 of ZZ. The four N-terminal residues of the H3 peptide are in direct contact with ZZ, while the rest of the tail is unstructured and solvent-exposed (Fig. 2b and Supplementary Fig. 2b). Ala1 of the H3 peptide inserts deeply into the negatively charged cavity of the protein, and its amino-terminal $NH_3^+$ group is restrained through interactions with ZZ, specifically the side-chain carboxyl groups of D824 and D848 and the carbonyl group of K822 (Fig. 2c).

Consistent with the structural results, blocking the amino-terminal $NH_3^+$ group of H3 Ala1 by an acetyl group or biotin impeded ZZZ3 ZZ-H3 interaction (Supplementary Fig. 2c); and deletion of the first two amino acids (Ala1–Arg2) of the H3 peptide completely abolished its interaction with all ZZ domains, as observed in peptide pulldown (Supplementary Fig. 1d) and NMR titration experiments (Supplementary Fig. 2d). Tryptophan fluorescence experiments revealed a $K_d$ of 29 μM for ZZZ3 ZZ association with the H3 peptide (Fig. 2d). Substitution of D824 or D848 in the ZZ domain with an alanine completely disrupted its interaction with the H3 peptide (Fig. 2d and Supplementary Fig. 2e, f). Importantly, all the H3-binding ZZ domains contain an invariable aspartate (D848 in ZZZ3) along with an aspartate (D824 in ZZZ3) or asparagine (in p300 and CBP ZZs) (Supplementary Fig. 2g and the accompanied manuscript), suggesting that recognition of Ala1 of H3 through the acidic cage is likely a common mechanism of the ZZ readers. However, simply introducing acidic residues into the non-H3-binding ZZ domains of TADA2A and TADA2B did not render them H3 binding activity (Supplementary Fig. 2h), suggesting that interactions other than with the acidic residues are also required. In line with this, an N-terminal Ala1 added to a non-H3-like peptide (AGSGSG) was insufficient to bind the ZZZ3 ZZ domain (Supplementary Fig. 2i).

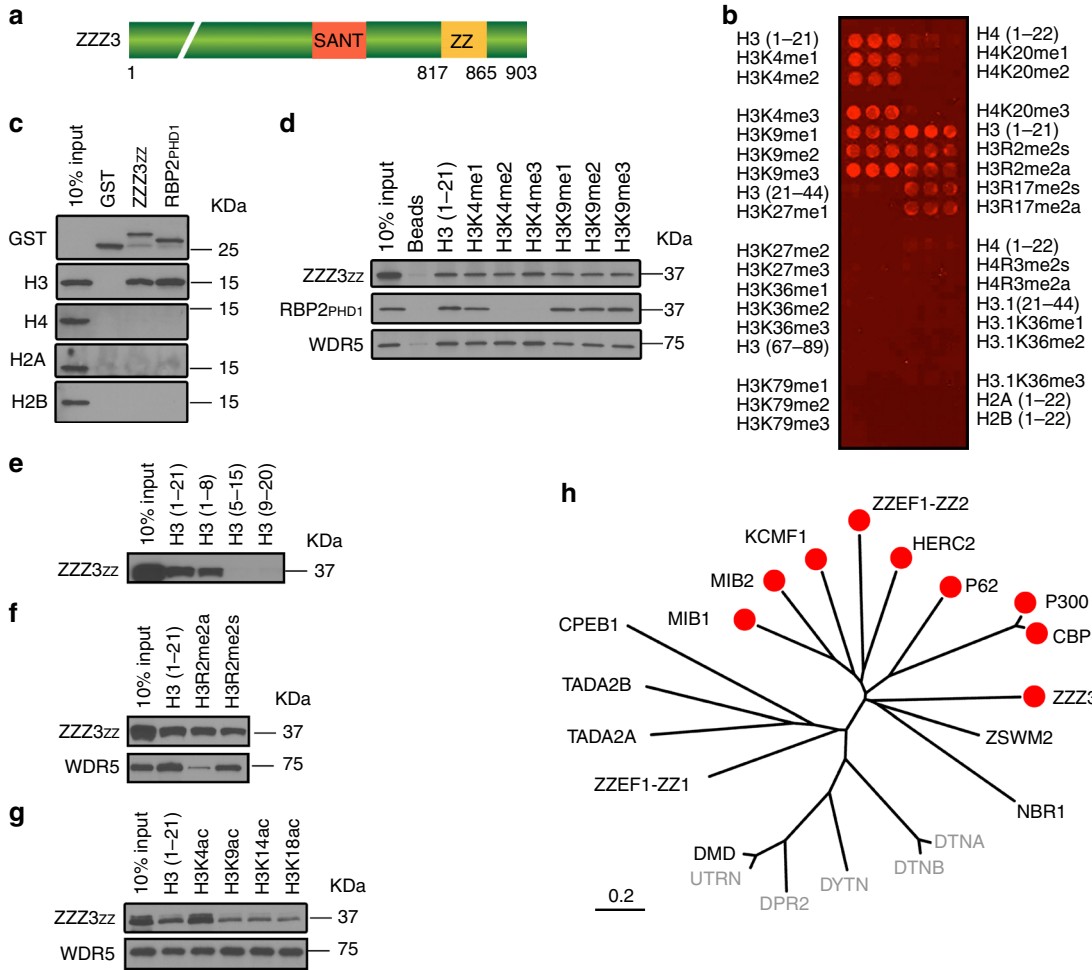

**Fig. 1** The ZZ domain of ZZZ3 recognizes histone H3 tail. **a** Schematic representation of ZZZ3 protein structure. The amino acid numbers of ZZ domain and the full-length protein are shown. **b** Histone peptide microarray probed with GST-ZZZ3 ZZ domain. Each peptide has three replicates. **c** Western blot analysis of GST pulldowns of GST-ZZZ3 ZZ with calf thymus histones. Pulldown of GST-RBP2 PHD1 is shown as a control. **d** Western blot analysis of peptide pull-down experiments of GST-ZZZ3 ZZ domain with the indicated histone H3 peptides. Peptide pulldowns of RBP2 PHD1 and WDR5 are shown for comparison. **e**–**g** Western blot analysis of peptide pulldown experiments of GST-ZZZ3 ZZ domain with the indicated H3 peptides. Peptide pulldowns of WDR5 are shown for comparison. **h** Phylogenetic tree presentation of the H3-binding activity of human ZZ proteins. The red dot indicates binding with H3 (Supplementary Fig. 1c–e). The cytoplasmic ZZ proteins (gray) are not tested in this study

**H3K4 acetylation moderately enhances the ZZZ3-H3 binding.** Although methylation has no apparent effect on the binding of ZZZ3 ZZ to the H3 peptide, acetylation on H3K4 moderately enhances the interaction, as observed in peptide arrays, pulldowns, and NMR titrations (Fig. 1b–d–g, Supplementary Figs 1b and 3a, b). Consistent with these observations, tryptophan fluorescence experiments showed an ~3-fold enhancement for ZZ association with the H3K4ac peptide ($K_d = 8.2\ \mu M$) over its association with the unmodified H3 peptide ($K_d = 29\ \mu M$) (Fig. 2d and Supplementary Fig. 3c). Microscale thermophoresis (MST) binding assays further corroborated the threefold enhancement (Fig. 2e and Supplementary Fig. 3d).

To understand the enhancement of binding to acetylated H3K4, we determined the solution structure of the ZZZ3 ZZ domain in complex with an H3K4ac peptide (aa 1–8) using NMR spectroscopy (Supplementary Fig. 3e and Supplementary Table 1). Structural comparison of the H3-bound and H3K4ac-bound ZZ revealed that H3K4 lays in a hydrophobic groove formed by F821, G820, and V819 in both complexes; however, the mean position of the acetylated side chain of Lys4 in the ensemble of the NMR structures is located closer to the side chain of F821 of ZZ compared to the mean position of the unmodified Lys4 (Fig. 2f

and Supplementary Fig. 3f). We speculated that the hydrophobic interaction between the neutral side chain of acetylated Lys4 and the aromatic ring of ZZ F821 might account for the increase in binding affinity. To test this hypothesis, we substituted F821 for an alanine and examined binding of the F821A mutant to the H3K4ac and unmodified H3 peptides using several orthogonal methods. We found that alanine substitution substantially decreases ZZ binding to either the unmodified or the Lys4-acetylated H3 peptides (Fig. 2g), and that the F821A mutant associates equally with both peptides in NMR titration and tryptophan fluorescence assays (Fig. 2d and Supplementary 3b–h). The acetylation-dependent enhancement of H3 binding by ZZ is reminiscent to the acetylation- or methylation-dependent enhancement in binding activity of other H3 readers. Much like the ZZZ3 ZZ domain, the DPF module of DPF3b or MORF binds ~3-fold tighter to H3K14ac than to the unmodified H3 peptide[20–23]. Structural comparison reveals that although the overall ZZ and DPF3b[23] exhibit distinct mechanisms for the recognition of the histone H3 tail, in both cases, the hydrophobic character of the acetyllysine-binding sites and the presence of aromatic residues most likely facilitate binding to a hydrophobic acetylated lysine species (Supplementary Fig. 4a, b).

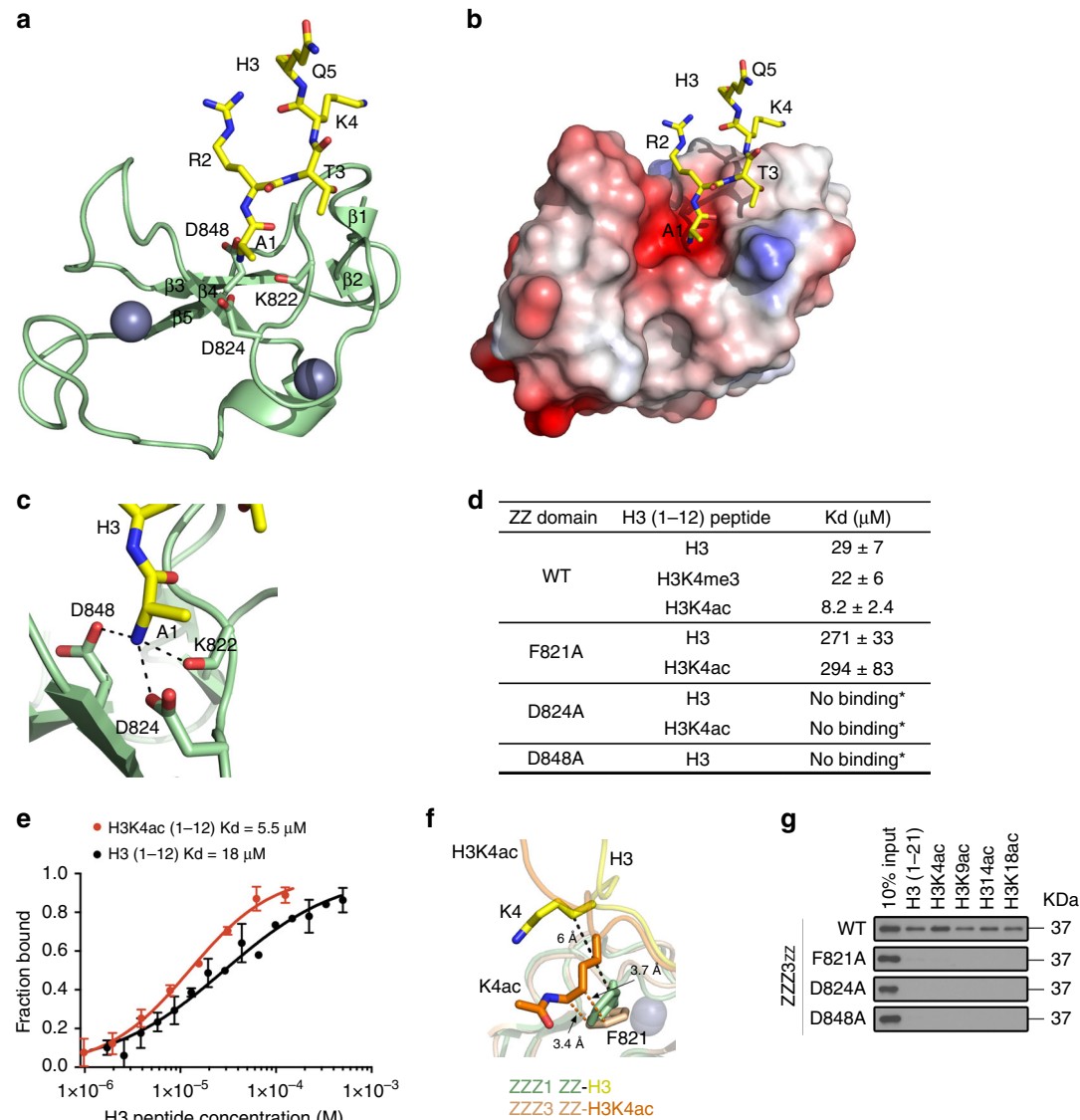

**Fig. 2** Structure of the ZZZ3 ZZ domain in complex with H3 peptides. **a** Ribbon diagram of the ZZZ3 $ZZ_{816-874}$ domain (green) in complex with the histone H3 peptide (yellow). **b** Electrostatic potential surface of ZZZ3 ZZ in complex with the histone H3 peptide (yellow stick). A positive electrostatic potential is colored in blue and negative in red. **c** Zoom-in view of the histone H3 Ala1 binding site. ZZZ3 ZZ domain is shown as a green ribbon and the histone H3 peptide as a yellow stick. The dash lines indicate short distances. **d** Binding affinities of WT ZZZ3 ZZ and mutants for the indicated histone peptides measured by tryptophan fluorescence. The experiments were carried out in triplicates. *: measured by NMR. **e** Binding curves used to determine the $K_d$ values by MST. **f** Zoom-in view of the mean position of Lys4 in each ensemble of 20 NMR structures of the complexes of ZZZ3 ZZ with unmodified H3 peptide and with H3K4ac peptide are superimposed. **g** Western blot analysis of histone peptide pulldowns of wild-type (WT) ZZZ3 ZZ and the indicated point mutants

Similarly, a single aromatic residue in the PHD finger of CHD4[24] provides favorable hydrophobic and cation-pi contacts with H3K9me3 (Supplementary Fig. 4c).

**ZZZ3 is critical for ATAC-dependent histone acetylation**. The catalytic subunit GCN5 or PCAF in the ATAC complex has been shown to catalyze acetylation of histone H3 on H3K9, H3K14, and H3K4[11,12,25]. Because ZZZ3 is a stoichiometric component of the ATAC complex, we speculated that recognition of H3 by the ZZZ3 ZZ domain may facilitate substrate association of ATAC, thus promoting H3 acetylation (Fig. 3a). To test this hypothesis, we carried out in vitro HAT assays of purified ATAC complex from cultured cells using histone peptides and reconstituted unmodified nucleosomes as substrates. We found that the purified ATAC complex shows robust acetylation activity on H3K9

and relatively weak activity on H3K4 (Fig. 3b, c). Importantly, deletion of the first two amino acids from histone H3 greatly reduced the HAT activity of ATAC on both the histone peptide and reconstituted nucleosomes. Compared with the unmodified H3 peptide, the Lys4-acetylated H3 peptide is a better substrate for ATAC (Supplementary Fig. 5a). Furthermore, substitution of F821 and D824 of ZZZ3 to an alanine substantially reduced the HAT activity of ATAC on H3 without affecting the complex integrity (Fig. 3d and Supplementary Fig. 5b, c). Together, these results suggest that binding of H3 by ZZZ3 is required for efficient H3 acetylation by the ATAC complex.

Consistent with the in vitro data, depletion of ZZZ3 by shRNA-mediated knockdown (KD) in H1299 and A549 lung adenocarcinoma cell lines resulted in reduction in global H3K9 and H3K4 acetylation levels (Fig. 3e). This effect is not due to the

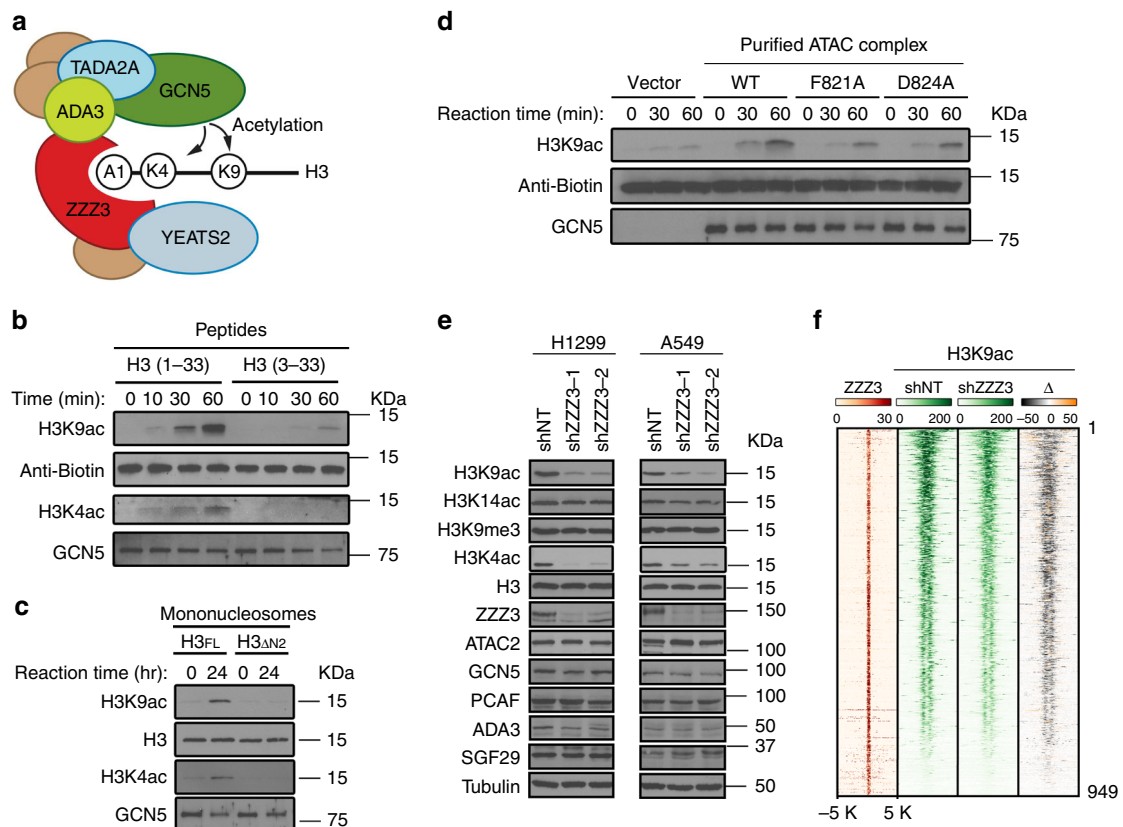

**Fig. 3** Recognition of H3 by ZZZ3 is critical for ATAC-dependent histone acetylation. **a** Schematic of ZZZ3 and the ATAC complex in recognition and acetylation of histone H3 tail. **b**, **c** Deletion the first two amino acids (Ala1–Arg2) of H3 abolishes the HAT activity of the ATAC complex in vitro. Western blot analysis of HAT assays of purified ATAC complex using the H3 (1–33) or H3 (3–33) peptides (**b**), or reconstituted mononucleosome containing full-length H3 or H3NΔ2 proteins (**c**) as substrates. Biotinylated peptides and H3 are shown as loading controls. **d** Point mutations of ZZ domain attenuate the HAT activity of the ATAC complex in vitro. Western blot analysis of HAT assays of purified ATAC complexes containing WT or mutant ZZZ3 using H3 (1–33) peptide as a substrate. Biotinylated peptides are shown as loading controls. **e** ZZZ3 KD reduces global H3K9ac and H3K4ac levels in cells. Western blot analysis of the indicated histone modifications and protein levels of ATAC subunits in control (shNT) and ZZZ3 KD (shZZZ3) cells. H3 and tubulin are used as loading controls. **f** Heatmap profiles of ZZZ3 in H1299 cells, H3K9ac in control (shNT) and ZZZ3 KD (shZZZ3) cells, and difference of H3K9ac (Δ, shZZZ3-shNT). The color key represents the signal density. Gray indicates a reduction in ZZZ3 KD cells compared to the control cells

loss of complex integrity as the protein levels of GCN5/PCAF and other ATAC complex components remained unchanged. ChIP experiments followed by high-throughput sequencing (ChIP-seq) revealed that the reduction of H3K9 acetylation is specific to ZZZ3-occupied genes (described below), as the H3K9ac levels on non-ZZZ3-occupied genes are largely not affected by ZZZ3 KD (Fig. 3f and Supplementary Fig. 5d). ChIP followed by quantitative real-time PCR (qPCR) analysis in ZZZ3 KD cells revealed that depletion of ZZZ3 reduces H3K9ac and H3K4ac levels on ZZZ3 target genes (Supplementary Fig. 6a–c). This reduction is likely due to the loss of ATAC occupancy on chromatin, as the binding of YEATS2, an ATAC-specific subunit, on ATAC target genes was substantially reduced upon ZZZ3 KD, whereas the occupancy of the SAGA subunit SPT20 and H3K9ac levels on SAGA target genes was not affected (Supplementary Fig. 6d–f). Depletion of either GCN5 or PCAF considerably reduced target gene H3K9 acetylation levels, and double KD showed an additive effect (Supplementary Fig. 6g, h), suggesting that likely both GCN5 and PCAF contribute to ATAC-dependent maintenance of histone acetylation.

**ZZZ3 is required for ATAC complex-dependent gene expression.**
To determine the functional importance of H3 recognition by the ZZ domain in cells, we first investigated its role in ZZZ3 binding to chromatin. We performed ChIP-seq experiments in H1299

cells to assess the genome-wide occupancy of ZZZ3 and its correlation with the ATAC complex and active transcription-associated histone marks (H3K4me3, H3K4ac, and H3K9ac). Using a validated ChIP-grade anti-ZZZ3 antibody[26], we identified 949 ZZZ3-occupied peaks that are strongly enriched in promoters (±3 kb of the transcription start site, TSS) (Supplementary Fig. 7a and Supplementary Data 1). We determined genome-wide occupancy of Flag-YEATS2 (Supplementary Data 1) by Flag ChIP-seq and used it as a surrogate of the ATAC complex. We found that although Flag ChIP-seq yielded many more peaks, the majority of the ZZZ3 peaks overlap with Flag-YEATS2 peaks (Supplementary Fig. 7b). Furthermore, more than three-quarters of ZZZ3 peaks co-localize with H3K4ac, H3K9ac, and H3K4me3 occupied peaks (Fig. 4a and Supplementary Fig. 7c), and all peaks are enriched at gene promoters (Fig. 4b and Supplementary Fig. 7d, e).

We then performed RNA-seq analysis in ZZZ3 KD cells to identify ZZZ3 regulated genes. We used two independent shRNAs targeting *ZZZ3* to minimize the off-target effect of shRNAs. We identified 1333 genes downregulated and 1836 genes upregulated in both shRNA KD cells compared with the cells treated with a non-targeting shRNA (Supplementary Fig. 8a–c and Supplementary Data 2), with the downregulated genes strongly enriched in the pathways of ribosome biogenesis (Supplementary Fig. 8d). Co-analysis of the ZZZ3 ChIP-seq and

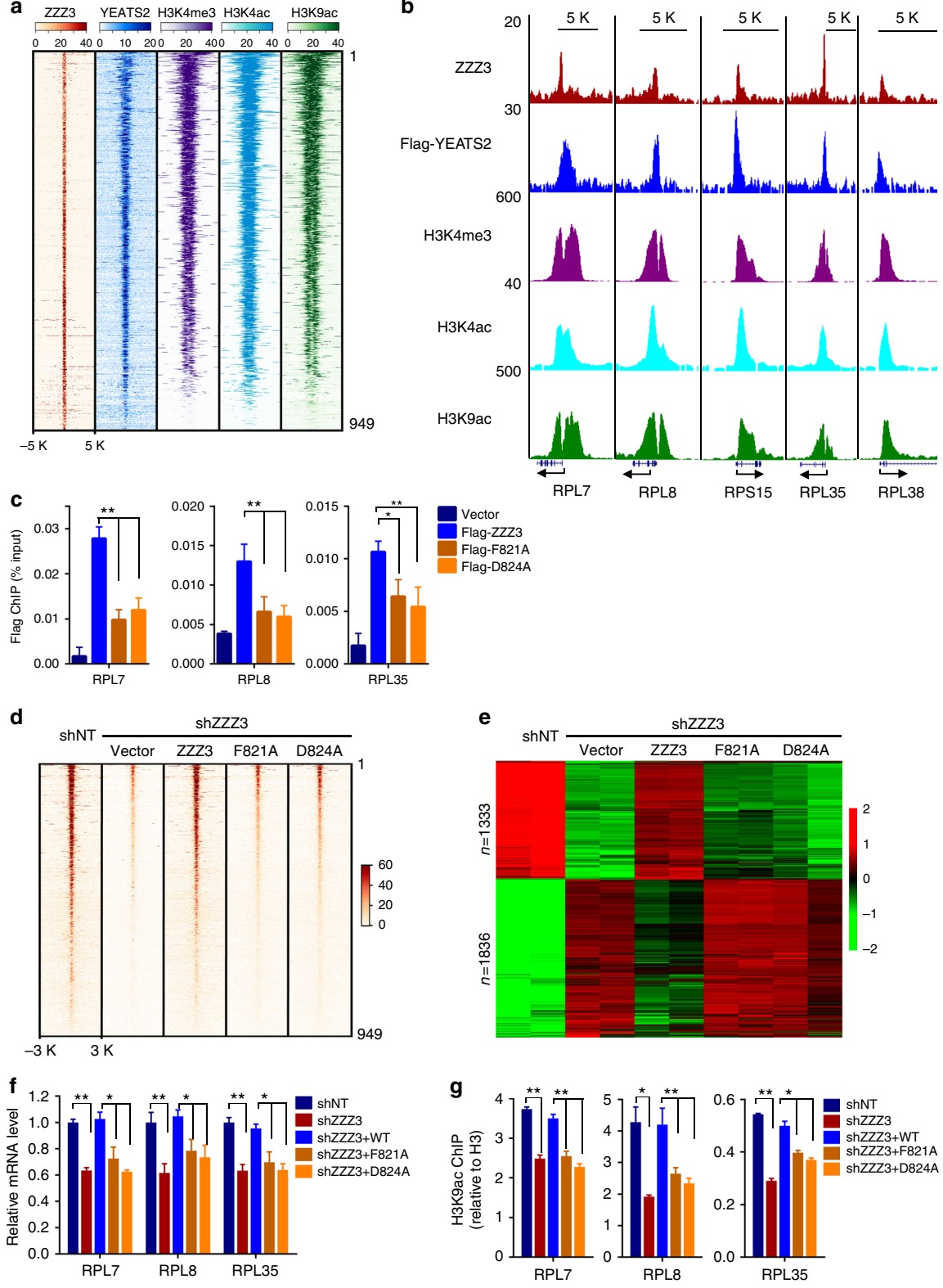

RNA-seq data uncovered 116 downregulated genes and 49 upregulated genes likely to be direct targets of ZZZ3 (Supplementary Fig. 9a and Supplementary Data 2). While the upregulated direct target genes are not significantly enriched in any pathway, the downregulated 116 genes, including 39 ribosomal protein genes, are again enriched in ribosome biogenesis (Supplementary Fig. 9b–d).

**The ZZ domain is critical for the function of ATAC.** Next, we asked whether recognition of H3 tail by the ZZ domain is required for ZZZ3 chromatin recruitment and function. To address this question, we performed "rescue" experiments by ectopically expressing Flag-tagged shRNA-resistant WT ZZZ3 or the H3-binding deficient mutants (F821A and D824A) in ZZZ3-depleted H1299 cells (Supplementary Fig. 10a, b). ChIP-seq and

**Fig. 4** ZZZ3 is required for ATAC-mediated H3K9 acetylation and gene activation. **a** Heatmaps of the ZZZ3, Flag-YEATS2, H3K4ac, H3K9ac, and H3K4me3 ChIP-seq signals centered on ZZZ3 binding sites in a ± 5-kb window. Loci were classified as ZZZ3 peaks enriched with active promoter marks and others without active promoter marks. The color key represents the signal density. **b** Representative genome-browser views of ZZZ3 (red), YEATS2 (blue), H3K4me3 (purple), H3K4ac (cyan), and H3K9ac (green) ChIP-seq signals. TSS is indicated by an arrow. **c** qPCR analysis of Flag-ZZZ3 ChIP on gene promoters in cells stably expressing Flag-tagged WT ZZZ3 or the indicated point mutants. **d** Heatmap of the normalized ZZZ3 ChIP-seq signal densities in control (shNT), ZZZ3 KD (shZZZ3) cells, and KD cells ectopically expressing shRNA-resistant WT ZZZ3 or the indicated mutants. The signal is centered on ZZZ3 binding site in a ± 3-kb window. **e** Heatmap representation of differentially expressed genes in cells as in **d** from two biological replicates of RNA-seq experiments. Fisher's exact test was used to define differentially expressed genes ($q < 0.01$). The color key represents normalized Log2 expression values. **f** qRT-PCR analysis of the expression of the indicated ribosomal protein genes in control (shNT), ZZZ3 KD (shZZZ3) cells, and KD cells ectopically expressing shRNA-resistant WT ZZZ3 or the indicated mutants. **g** qPCR analysis of H3K9ac ChIP on a promoter of the indicated ribosomal protein genes in cells as in **f**. Error bars represent s.e.m. of three biological replicates. *$p < 0.05$; **$p < 0.01$ (two-tailed unpaired Student's $t$-test)

ChIP-qPCR using the anti-Flag M2 antibody revealed that substituting alanine for either F821 or D824 resulted in reduced ZZZ3 occupancy at target gene promoters (Fig. 4c, Supplementary Fig. 10c, d, and Supplementary Data 1). ChIP-seq experiments using the ZZZ3-specific antibody demonstrated that introducing WT ZZZ3, but not the F821A or D824A mutant, into the ZZZ3 KD cells restored genomic distribution of ZZZ3 (Fig. 4d, Supplementary Fig. 10e and Supplementary Data 1). Finally, transcriptional profiling analyses revealed that the H3-binding deficient mutants failed to rescue global transcriptional changes of the ZZZ3 KD cells (Fig. 4e and Supplementary Data 2). The failure of the H3-binding deficient mutants of ZZZ3 in restoring target gene expression (Fig. 4f) was likely attributed to the defect of the ZZZ3 mutants associated ATAC complex in acetylating histones on target gene promoters (Fig. 4g).

In summary, we identified a new reader of histone H3 with a recognition mode distinct from all known histone H3 readers. The ZZ-H3 interaction is important for the recruitment of ZZZ3 and the ATAC complex to maintain an acetylated, open chromatin environment, thus promoting gene activation. Notably, TADA2A and TADA2B components of the ATAC and SAGA complexes, respectively, also contain a ZZ module, but do not bind to histone H3. In future studies, it will be interesting to determine the functions of these ZZ domains and how they have evolved to perform functions distinct from recognizing histone H3.

## Methods

**Materials**. Human ZZZ3 cDNA (NCBI Gene ID 26009) was cloned into pENTR3C, and subsequently cloned into p3FLAG, pCDH-Flag destination vectors using Gateway techniques (Invitrogen). The cDNA encoding the ZZ domain (amino acids 800–903) of human ZZZ3 was cloned into the pGEX-6P1 vector (Novagen). Point mutations were generated using a site-directed mutagenesis kit (Stratagene). Histone peptides bearing different modifications were synthesized at the CPC and Scilight Biotechnology. Purified recombinant mononucleosomes were purchased from EpiCypher, Inc. The anti-SGF29 and ChIP-seq-grade ZZZ3 antibodies were provided by Dr. Laszlo Tora[26]. Anti-histone antibodies (with dilution for western blotting) including anti-H3 (Ab1791, 1:20,000), anti-H3K9ac (Ab32129, 1:1000), anti-H3K14ac (Ab52946, 1:1000), anti-H4 (Ab7311, 1:1000), anti-H2A (Ab18255, 1:1000), anti-H2B (Ab52599, 1:1000), and anti-H3K9me3 (Ab8898, 1:1000) antibodies were obtained from Abcam. Anti-GCN5 (sc-20698, 1:2000), anti-PCAF (sc-13124, 1:200), anti-ADA3 (sc-98821, 1:1000), anti-ATAC2 (sc-398475, 1:1000), and anti-GST (sc-459, 1:1000) antibodies were from Santa Cruz. Anti-ZZZ3 (SAB4501106, 1:1000), anti-Flag (M2, 1:5000), and anti-tubulin (T8328, 1:5000) antibodies were from Sigma. Anti-H3K9ac (61251) and anti-H3K4ac (39381) for ChIP-seq were from Active Motif. The shRNAs targeting human ZZZ3 were obtained from Sigma. shRNA sequences were ZZZ3#1, 5′-AT CGTATGGGACCAATATAC; and ZZZ3#2, 5′-GCATCAGATGACGAAAGTA TT.

**Peptide microarray, peptide, and GST pulldown assays**. Peptide microarray and peptide pulldown assays were performed as described previously[27]. Briefly, biotinylated histone peptides were printed in triplicate onto a streptavidin-coated slide (PolyAn) using a VersArray Compact Microarrayer (Bio-Rad). After a short blocking with biotin (Sigma), the slides were incubated with the GST-fused ZZZ3 ZZ or other desired proteins in binding buffer (50 mM Tris–HCl 7.5, 300 mM NaCl, 0.1% NP-40, 1 mM PMSF, and 20% fetal bovine serum) overnight at 4 °C

with gentle agitation. After being washed with the same buffer, the slides were probed with an anti-GST primary antibody and then a fluorescein-conjugated secondary antibody and visualized using a GenePix 4000 scanner (Molecular Devices).

For the peptide pulldown assays, 1 μg of biotinylated histone peptides with different modifications were incubated with 1–2 μg of GST-fused proteins in binding buffer (50 mM Tris–HCl 7.5, 300 mM NaCl, 0.1% NP-40, and 1 mM PMSF) overnight. Streptavidin beads (Amersham) were added to the mixture, and the mixture was incubated for 1 h with rotation. The beads were then washed three times and analyzed using SDS-PAGE and western blotting.

For GST pull-down experiments, 2 μg of GST-fused proteins were incubated with 10 μg of calf thymus total histones (Worthington) in binding buffer (50 mM Tris–HCl 7.5, 1 M NaCl, 1% NP-40, 0.5 mM EDTA, and 1 mM PMSF plus protease inhibitors (Roche)) at 4 °C overnight, followed by an additional 1 h of Glutathione-Sepharose beads (Amersham) incubation. The beads were then washed five times and analyzed using SDS-PAGE and western blotting. Other steps are essentially the same as described above.

**Protein expression and purification**. The human ZZZ3 ZZ domain (aa 816–874 and 800–903) was cloned into a pGEX 6p-1 vector and expressed in BL21 (DE3) RIL cells. Protein production was induced with 0.2 mM IPTG and incubated overnight at 18 °C in Luria broth (LB) or minimal media (M9) supplemented with 0.05 mM ZnCl₂. For the production of $^{13}$C/$^{15}$N-labeled or $^{15}$N-labeled samples, $^{15}$NH₄Cl and $^{13}$C₆-glucose were used in the M9 medium. The GST-ZZ proteins were purified on glutathione Sepharose 4B beads (GE Healthcare) in 20 mM Tris–HCl (pH 7.0) buffer, supplemented with 100 mM NaCl and 5 mM DTT. The GST tag was cleaved overnight at 4 °C with PreScission protease. Unlabeled and $^{13}$C/$^{15}$N-labeled proteins were further purified by size-exclusion chromatography and concentrated in Millipore concentrators. All mutants were generated by site-directed mutagenesis using the Stratagene QuikChange mutagenesis protocol, grown, and purified as WT proteins.

**NMR experiments**. NMR experiments were carried out at 298 K on Varian INOVA 600- and 900-MHz spectrometers. NMR samples contained 20 mM Tris–HCl (pH 7.0) buffer, supplemented with 100 mM NaCl, 5 mM DTT, and 8% D2O. The chemical shift assignments for apo-state were obtained by a set of triple-resonance experiments[28] with nonlinear sampling[29] using a $^{13}$C/$^{15}$N-labeled ZZZ3₈₁₆–₈₇₄ sample. To obtain the H3- or H3K4ac-bound state, 10 mM H3 (1–12) peptide or 4 mM H3K4ac (1–8) was added to 2 mM $^{13}$C/$^{15}$N-labeled ZZZ3₈₁₆–₈₇₄ sample. The concentration of peptides was calculated by weighing ~1 mg of each peptide and dissolving in water to obtain 50–100 mM stock solutions. The stock solutions were prepared independently for the NMR and fluorescence experiments. Triple-resonance experiments were collected to confirm the assignments for the complex state. 3D $^{15}$N- and $^{13}$C-edited NOESY–HSQC spectra (mixing time of 100 ms) were collected to obtain distance restraints within ZZZ3-ZZ. 2D $^{15}$N-/$^{13}$C-filtered NOESY and TOCSY–HSQC spectra were collected to obtain the chemical shift assignment for the H3 peptide. 3D $^{15}$N-/$^{13}$C-filtered $^{13}$C- and $^{15}$N-edited NOESY–HSQC spectra (mixing time of 200 and 300 ms for H3 (1–12), and of 200 ms for H3K4ac (1–8)) were collected to obtain the intermolecular distance restraints.

In total, 0.1 mM uniformly $^{15}$N-labeled WT or mutated ZZZ3-ZZ was used in NMR titrations. Binding was characterized by monitoring chemical shift changes in the proteins induced by histone peptides (synthesized by SynPeptide). Dissociation constant ($K_d$) was determined by applying PCA to the $^1$H,$^{15}$N HSQC titration spectra in TREND[30]. Each binding isotherm was fitted using the following equation:

$$p_{bound} = ||PC1|| = \frac{\left(([L] + [P] + K_d) - \sqrt{([L] + [P] + K_d)^2 - 4[P][L]}\right)}{2[P]}$$

where $[L]$ is the concentration of the histone peptide, $[P]$ is the concentration of ZZ, $p_{bound}$ is the fraction of protein bound to a ligand, and $PC1$ is the normalized

principal component, obtained by TREND, that indicates the change in the population of the bound state. The errors of the $K_d$ value are the fitting uncertainties from nonlinear least-squares fits in Kaleidagraph.

**Structure determination for the ZZZ3-H3 complex**. Calculation of the structure of ZZZ3-ZZ (aa 816–874) in complex with H3 (1–12) peptide or H3K4ac (1–8) peptide was carried out using interproton NOE-derived distance restraints and dihedral angle restraints. Spectra were processed and analyzed with NMRDraw and CcpNmr Suite[31]. The program DANGLE in CcpNmr Suite was used to predict dihedral angles $\psi$ and $\varphi$ restraints. Hydrogen bonds were derived from characteristic NOE patterns in combination with dihedral angles. The structures were calculated initially with XPLOR-NIH and refined with AMBER[32,33]. Hundred structures were calculated, and the ensemble of 20 conformers with the lowest total energy was selected to represent the complex of ZZZ3-ZZ and H3 peptides. The quality of the structures was validated using the program PROCHECK-NMR. The structural statistics is listed in Supplementary Table 1.

**Fluorescence spectroscopy**. Spectra were recorded at 25 °C on a Fluoromax-3 spectrofluorometer (HORIBA). The samples containing 1.0 μM ZZZ3 ZZ (aa 800–903) and progressively increasing concentrations of the histone peptide were excited at 295 nm. Experiments were performed in buffer containing 20 mM Tris–HCl (pH 7.2), 150 mM NaCl, and 1 mM DTT, with the exception of experiments shown in Supplementary Fig. 3b, which were performed in 100 mM NaCl buffer. Emission spectra were recorded over a range of wavelengths between 330 and 360 nm with a 0.5-nm step size and a 1-s integration time and averaged over three scans. The $K_d$ values were determined using a nonlinear least-squares analysis and the equation:

$$\Delta I = \Delta I_{max} \frac{\left( ([L] + [P] + K_d) - \sqrt{([L] + [P] + K_d)^2 - 4[P][L]} \right)}{2[P]}$$

where $[L]$ is the concentration of the histone peptide, $[P]$ is the concentration of ZZZ3 ZZ domain, $\Delta I$ is the observed change of signal intensity, and $\Delta I_{max}$ is the difference in signal intensity of the free and bound states of the ZZ domain. The $K_d$ value was averaged over three separate experiments, with error calculated as the standard deviation between the runs.

**Microscale thermophoresis (MST) binding assay**. The MST experiments were performed using a Monolith NT.115 instrument (NanoTemper) as described previously[34]. All experiments were performed with the purified ZZ domain (a.a. 800–903) in a buffer containing 20 mM Tris–HCl (pH 7.0), 150 mM NaCl, and 3 mM DTT. The final concentration of the C-terminal fluorescein-labeled histone H3 peptide (1–12, KE BIOCHEM) was kept at 80 nM. Dissociation constants for the interaction between ZZ with unlabeled peptides H3 (1–12) and H3K4ac (1–12) were measured using a displacement assay in which increasing amounts of unlabeled peptides were added into a preformed ZZ:H3-FAM complex prepared by supplementing 5 μM ZZ into each sample. The measurements were performed at 50% LED and 40% MST power with 3-s laser on time and 22 s off time. For all measurements, samples were loaded into premium capillaries and 1400–1700 counts were obtained for the fluorescence intensity. The $K_d$ and $IC_{50}$ values were determined with the MO.Affinity Analysis software (NanoTemper Technologies GmbH), using two independent MST measurements. The $K_i$ values for unlabeled peptides with ZZ were determined from the $IC_{50}$ values observed in the displacement assay and converted by the following equation:

$$K_i = [I]_{50} \bigg/ \left( \frac{[L]_{50}}{K_d} + \frac{[P]_0}{K_d} + 1 \right)$$

where $[I]_{50}$ is the concentration of free unlabeled ligand at 50% binding and $[L]_{50}$ is the concentration of free labeled H3 peptide at 50% binding. The $K_d$ value is the dissociation constant of labeled H3 peptide determined in the direct binding experiment described above. Measurements for H3 unmodified and H3K4ac peptides were done in triplicates and duplicates, respectively.

**Affinity purification and HAT assays**. Nuclear extracts were prepared from stable cells using a standard protocol[35]. Briefly, approximately 2 mg of nuclear extract was incubated with 25 μl of M2-agarose beads (Sigma) overnight at 4 °C. The beads were washed four times with high-salt wash buffer (10 mM HEPES, pH 7.9, 25% glycerol, 1.5 mM $MgCl_2$, 300 mM KCl, and 0.1% Triton X-100), followed by two washes with low-salt wash buffer (10 mM HEPES, pH 7.9, 25% glycerol, 1.5 mM $MgCl_2$, 100 mM KCl, and 0.1% Triton X-100). Elution was achieved by two consecutive incubations of the beads with 0.5 mg/ml triple-FLAG peptide (Sigma) in 200 μl of low-salt wash buffer. All buffers contained phenylmethylsulfonyl fluoride and protease inhibitor cocktail (Roche).

Purified complexes were incubated for indicated times at 37 °C with peptides (1 μg) or mononucleosomes (10 μg) and 0.1 mM acetyl-CoA in HAT reaction buffer (50 mM Tris, pH 8.0, 0.1 mM EDTA, 10% glycerol, 1 mM PMSF, and 1 mM DTT) in a total volume of 50 μL. Reactions were quenched by flash-freezing in

liquid nitrogen and then analyzed by SDS-PAGE and western blot. Uncropped western blots are shown in Supplementary Figure 11.

**Cell culture and viral transduction**. All cell lines were validated by STR DNA fingerprinting performed by the MDACC CCSG-funded Characterized Cell Line Core (NCI # CA016672). Human HEK 293 T (ATCC) cells were maintained in DMEM (Cellgro) supplemented with 10% fetal bovine serum (Sigma). Human lung cancer cell lines H1299 and A549 were cultured in RPMI 1640 (Cellgro) supplemented with 10% fetal bovine serum. Lentiviral transduction was performed as described previously[27]. Briefly, 293T cells were co-transfected with pMD2.G, pPAX2 (Addgene) and pLKO shRNA, or pCDH cDNA constructs and control vectors, and then viral supernatants were collected after 48 h. For infections, cells were incubated with viral supernatants in the presence of 8 μg/ml polybrene. After 48 h, the infected cells were selected with puromycin (2 μg/ml) for pLKO clones or blasticidin (10 μg/ml) for pCDH clones for 3–4 days before experiments.

**ChIP and ChIP-seq analysis**. ChIP analysis was performed essentially as described previously[27]. Briefly, cells were cross-linked with 1% formaldehyde for 10 min and stopped with 125 mM glycine. Nuclei were isolated by re-suspending the cells in swelling buffer containing 5 mM PIPES, pH 8.0, 85 mM KCl, 1% NP-40, and a complete protease inhibitor for 20 min at 4 °C. The isolated nuclei were re-suspended in nuclei lysis buffer (50 mM Tris, pH 8.0, 10 mM EDTA, and 1% SDS) and sonicated using a Bioruptor Sonicator (Diagenode). The samples were immunoprecipitated with 2–4 μg of the appropriate antibodies overnight at 4 °C. Protein A/G beads were added and incubated for 1 h, and the immunoprecipitates were washed twice each with low-salt, high-salt, and LiCl buffers. Eluted DNA was reverse-cross-linked, purified using PCR purification kit (Qiagene), and analyzed by quantitative real-time PCR on the ABI 7500-FAST System using the Power SYBR Green PCR Master Mix (Applied Biosystems). Statistical differences were calculated using a two-way unpaired Student's $t$-test. The primers used for qPCR are listed in Supplementary Data 3.

For ChIP-seq, ChIP experiments were carried out essentially the same as described above. Samples were sequenced using the Illumina Solexa Hiseq 2500. The raw reads were mapped to human reference genome NCBI 37 (hg19) by Solexa data-processing pipeline, allowing up to two mismatches. The genome ChIP-seq profiles were generated using MACS 1.3.6 with only unique mapped reads. Clonal reads were automatically removed by MACS. The ChIP-seq profiles were normalized to 10,000,000 total tag numbers, and peaks were called at $p$-values $\leq$ 1e-8 with input DNA as control. The input DNA was also used for peak calling. Nonspecific peaks called in the input DNA were used to remove false-positive peaks from ChIP samples. For ChIP-seq peak calling, promoters are defined as [−3k ~ 3k] bp around TSSs of RefSeq genes. A binomial test was used to evaluate the significance of ChIP-seq peak occupancy (at least 1nt) on promoters. ChIP-seq heatmap was drawn by the seqplots R package (http://github.com/przemol/seqplots).

**RNA extraction, real-time PCR, and RNA-seq analysis**. Reverse transcription PCR and real-time PCR were performed as previously described[27]. Total RNA was extracted using an RNeasy plus kit (Qiagen) and reverse-transcribed using an iScrip reverse transcription kit (Bio-Rad). Quantitative real-time PCR (qPCR) analyses were performed as described previously using Power SYBR Green PCR Master Mix and the ABI 7500-FAST Sequence Detection System (Applied Biosystems). Gene expressions were calculated following normalization to GAPDH levels using the comparative Ct (cycle threshold) method. For glucose starvation, gene expressions were calculated following normalization to cell number using the comparative Ct (cycle threshold) method. Statistical differences were calculated using a two-way unpaired Student's $t$-test. The primer sequences for qPCR are listed in Supplementary Data 3.

RNA-seq samples were sequenced using the Illumina Hiseq 2500, and raw reads were mapped to the human reference genome (hg19) and transcriptome using Tophat 2.1.0 (http://ccb.jhu.edu/software/tophat/index.shtml). Read counts for each transcript were calculated using HTseq v0.6.1 using default parameters. Differential gene expression analyses were performed using the "exactTest" function in edgeR v3.0. Gene Ontology analysis was performed using the DAVID Bioinformatics Resource 6.7. The gene expression heatmap was generated using pheatmap package in CRAN (https://cran.r-project.org/package=pheatmap). The volcano plot was drawn using ggplot2 package in R computing environment (https://cran.r-project.org/package=ggplot2).

**Statistical analyses**. Experimental data are presented as mean ± s.e.m. unless stated otherwise. Statistical significance was calculated by two-tailed unpaired $t$-test on two experimental conditions with $p < 0.05$ considered statistically significant unless stated otherwise. Statistical significance levels are denoted as follows: *$p < 0.05$; **$p < 0.01$; ***$p < 0.001$; n. s.: not significant. No statistical methods were used to predetermine sample size. For overlap analysis, a hypergeometric test was performed for a two-way Venn diagram by the R software unless otherwise stated. Super exact test was performed for the three-way Venn diagram by the R package (https://lib.ugent.be/CRAN/web/packages/SuperExactTest).

## Data availability

Structure data have been deposited in Protein Data Bank under accession numbers 6E83 and 6E86. The ChIP-seq and RNA-seq data have been deposited in the Gene Expression Omnibus database under accession number GSE100009.

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

## Acknowledgements

We thank M. Bedford, S.Y.R. Dent, B. Gan, J. Kurie, and H. Lee for sharing reagents and scientific discussions. We thank B. Dennehey for editing the manuscript. We thank the MD Anderson Science Park Next-Generation Sequencing Facility (CPRIT RP120348) for Solexa sequencing. This work was supported in part by grants from NIH/NCI (CA204020), Cancer Prevention and Research Institute of Texas (RP160237 and RP160739), and Welch Foundation (G1719) to X.S., NIH (GM106416, GM125195, and GM100907) to T.G.K., and CPRIT (RP150292) and NIH (HG007538 and CA193466) to W.L. X.S. is a Leukemia & Lymphoma Society Career Development Program Scholar.

## Author contributions

W.M., Y.Z., and J.L. contributed equally to this work. X.S., T.K., W.M., and Y.Z. conceived the study. W.M. performed the biochemical and cellular studies; Y.Z. performed the structural, NMR, and fluorescence studies with help from Q.T. and A.H.T; J.L. performed bioinformatics analysis; X.W., D.P., and Y.X provided technical assistance; X.S., T.K., W.M., and Y.Z. wrote the paper with comments from H.W., J.L., and W.L.

## Additional information

**Competing interests:** X.S. is a Scientific Advisory Board member of EpiCypher. All other authors declare no competing interests.

