## [Peer Review File · Nature Communications]

REVIEWERS' COMMENTS:

Reviewer #3 (Remarks to the Author):

This is a really nice manuscript at present, and the findings presented represent an important advance for the field. The majority of my concerns were adequately addressed.

Here are lingering issues that I feel can be addressed without another round of review:

1.) It is still not stated how peptides used in the NMR/trp fluorescence experiments were quantified in the methods. (I mentioned in previous review that peptide mass is not a tremendously accurate way to evaluate concentration as residual solvent/salt, static electricity, and balance precision preclude such accuracy.) The new MST experiments add additional support for this point, minimizing the impact of this point, as does the revision to focus on H3 binding and function, as distinct from the prior overemphasis of the potential H3K4ac binding enhancement.

2.) The shRNA data is still pretty mismatched in terms of genes affected, so are decent, but perhaps not as clean an experiment as would be ideal (see prior suggestion for CRISPR editing). The data presented in the rebuttal figure are not very compelling, R3a makes my prior point about the modest overlap of affected genes which should approach unity if there were not substantial off-target effects. Essentially the interpretation is there are more off-target effects than correlated on-target ones, which is not a great perturbation with which to robustly interpret on-target biology. R3b is dominated by genes that are minimally perturbed in both cases, so such an analysis is a suboptimal way to evaluate targeting similarity of shRNA. The heatmaps perhaps indicate rescue, but Venn analysis would be more helpful to evaluate the rescue argument and should be included in the revision. This remains a weak part of the paper, but in the context of voluminous other robust data, I am comfortable with it.

REVIEWERS' COMMENTS:

Reviewer #3 (Remarks to the Author):

This is a really nice manuscript at present, and the findings presented represent an important advance for the field. The majority of my concerns were adequately addressed.

Here are lingering issues that I feel can be addressed without another round of review:

1.) It is still not stated how peptides used in the NMR/trp fluorescence experiments were quantified in the methods. (I mentioned in previous review that peptide mass is not a tremendously accurate way to evaluate concentration as residual solvent/salt, static electricity, and balance precision preclude such accuracy.) The new MST experiments add additional support for this point, minimizing the impact of this point, as does the revision to focus on H3 binding and function, as distinct from the prior overemphasis of the potential H3K4ac binding enhancement.

Answer: The following sentences of peptide quantification are added to Methods in the NMR subsection: Concentration of peptides was calculated by weighting ~1 mg of each peptide and dissolving in water to obtain 50-100 mM stock solutions. The stock solutions were prepared independently for the NMR and fluorescence experiments.

2.) The shRNA data is still pretty mismatched in terms of genes affected, so are decent, but perhaps not as clean an experiment as would be ideal (see prior suggestion for CRISPR editing). The data presented in the rebuttal figure are not very compelling, R3a makes my prior point about the modest overlap of affected genes which should approach unity if there were not substantial off-target effects. Essentially the interpretation is the there are more off-target effects than correlated on-target ones, which is not a great perturbation with which to robustly interpret on-target biology. R3b is dominated by genes that are minimally perturbed in both cases, so such an analysis is a suboptimal way to evaluate targeting similarity of shRNA. The heatmaps perhaps indicate rescue, but Venn analysis would be more helpful to evaluate the rescue argument and should be included in the revision. This remains a weak part of the paper, but in the context of voluminous other robust data, I am comfortable with it.

Answer: As suggested, Venn diagram of overlap analysis of down regulated genes in shZZZ3-1 KD cells and gene rescued by WT ZZZ3 is shown in Supplementary Fig. 8c.